# GLOBAL PROTOTYPE ENCODING FOR INCREMENTAL VIDEO HIGHLIGHTS DETECTION

## ABSTRACT

Video highlights detection (VHD) is an active research field in computer vision, aiming to locate the most user-appealing clips given raw video inputs. However, most VHD methods are based on the closed world assumption, *i.e.*, a fixed number of highlight categories is defined in advance and all training data are available beforehand. Consequently, existing methods have poor scalability with respect to increasing highlight domains and training data. To address above issues, we propose a novel video highlights detection method named **G**lobal **P**rototype **E**ncoding (GPE) to learn incrementally for adapting to new domains via parameterized prototypes. To facilitate this new research direction, we collect a finely annotated dataset termed *LiveFood*, including over 5,100 live gourmet videos that consist of four domains: *cooking*, *eating*, *ingredients* and *presentation*. To the best of our knowledge, this is the first work to explore video highlights detection in the incremental learning setting, opening up new land to apply VHD for practical scenarios where both the concerned highlight domains and training data increase over time. We demonstrate the effectiveness of GPE through extensive experiments. Notably, GPE surpasses popular domain-incremental learning methods on *LiveFood*, achieving significant mAP improvements on all domains. The code and dataset will be made publicly available.

## 1 INTRODUCTION

The popularization of portable devices with cameras greatly promotes the creation and broadcasting of online videos. These sufficient video data serve as essential prerequisites for relevant researches, *e.g.* video summarization (Potapov et al., 2014; Song et al., 2015; Zhang et al., 2018; Fajtl et al., 2018; Zhu et al., 2021), video highlights detection (VHD) (Yang et al., 2015; Xiong et al., 2019; Lei et al., 2021; Bhattacharya et al., 2021), and moment localization (Liu et al., 2018; Zhang et al., 2020; Rodriguez et al., 2020), to name a few. Currently, most VHD methods are developed under the closed world assumption, which requires both the number of highlight domains and the size of training data to be fixed in advance. However, as stated in Rebuffi et al. (2017), natural vision systems are inherently incremental by consistently receiving new data from different domains or categories. Taking the gourmet video as an example, in the beginning, one may be attracted by the clips of eating foods, but lately, he/she may raise new interests in cooking and want to checkout the detailed cooking steps in the same video. This indicates that the target set the model needs to handle is flexible in the open world. Under this practical setting, all existing VHD methods suffer from the scalability issue: they are unable to predict both the old and the newly added domains, unless they retrain models on the complete dataset. Since the training cost on videos is prohibitive, it is thus imperative to develop new methods to deal with the above incremental learning issues.

Broadly speaking, there exist two major obstacles that hinder the development of incremental VHD: a high-quality VHD dataset with domain annotations and strong models tailored for this task. Recall existing datasets that are widely used in VHD research, including SumMe (Gygli et al., 2014), TVSum (Song et al., 2015), Video2GIF (Gygli et al., 2016), PHD (Garcia del Molino & Gygli, 2018), and QVHighlights (Lei et al., 2021), all of them suffer from threefold drawbacks: (1) only the feature representations of video frames are accessible instead of the raw videos, thus restricting the application of more powerful end-to-end models; (2) most datasets only have a limited number of videos with short duration and coarse annotations, which are insufficient for training deep models; (3) none of them has the video highlight domain or category labels, thus can not be directly used in

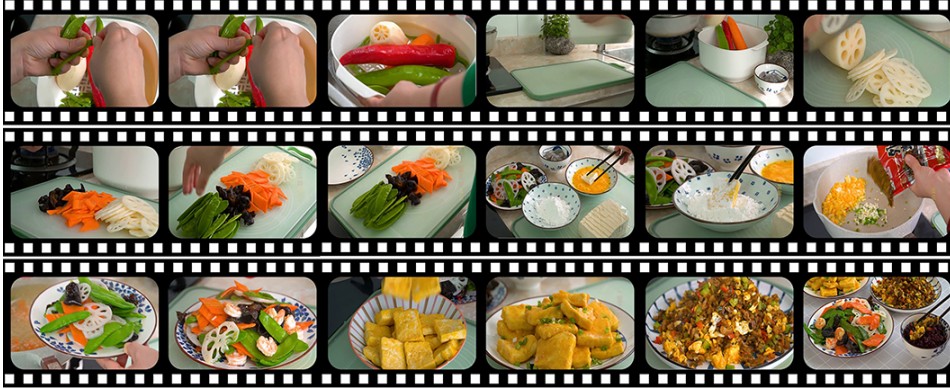

Figure 1: The *LiveFood* dataset. The row from top to bottom illustrates examples of vanilla clips, *ingredients* and *presentation* respectively. More samples are attached in Appendix A.2.

incremental learning. In order to bridge the gap between VHD and incremental learning, we first collect a high-quality gourmet dataset from live videos, namely *LiveFood*. It contains over 5,100 carefully selected videos with 197 hours in total. Four domains are finely annotated, *i.e., cooking*, *eating*, *ingredients* and *presentation*. These related but distinctive domains provide a new test bed for incremental VHD tasks.

To solve this new task, we propose a competitive model: **G**lobal **P**rototype **E**ncoding (GPE) to learn new highlight concepts incrementally while still retaining knowledge learned in previous video domains/data. Specifically, GPE first extracts frame-wise features using a CNN, then employs a transformer encoder to aggregate the temporal context to each frame feature, obtaining temporal-aware representations. Furthermore, each frame is classified by two groups of learnable prototypes: highlight prototypes and vanilla prototypes. With these prototypes, GPE optimizes a distance-based classification loss under $L_2$ metric and encourages incremental learning by confining the learned prototypes in new domains to be close to that previously observed. We systematically compare the GPE with different incremental learning methods on *LiveFood*. Experimental results show that GPE outperforms other methods on highlight detection accuracy (mAP) with much better training efficiency, using no complex exemplar selection or complicated replay schemes, strongly evidencing the effectiveness of GPE.

The main contributions of this paper are summarized as follows:

- We introduce a new task named incremental video highlights detection, which has important applications in practical scenarios. A high-quality *LiveFood* dataset is collected to facilitate research in this direction. *LiveFood* comprises over 5,100 carefully selected gourmet videos in high resolution, providing a new test bed for video highlights detection and domain-incremental learning tasks.

- We propose a novel end-to-end model for solving incremental VHD, *i.e.,* **G**lobal **P**rototype **E**ncoding (GPE). GPE can incrementally identify highlight and vanilla frames in new highlight domains via learning extensible and parameterized highlight/vanilla prototypes. GPE achieves superior performance compared with other incremental learning methods, improving the detection performance (mAP) by 1.57% on average. The above results suggest that GPE can serve as a strong baseline for future research.

- We provide comprehensive analyses of *LiveFood* as well as the proposed GPE model for deepening the understanding of both, as well as giving helpful insight for future development. We hope our work can inspire more researchers to work in incremental VHD, finally pushing forward the application of VHD in practical scenarios.

## 2    RELATED WORK

**Video Highlights Detection** (VHD) is an important task in video-related problems. This line of research can be roughly divided into two groups, namely the ranking-based and regression-based

methods. Yao et al. (2016) employs a ranking model to learn the relationship between highlights and non-highlights, assigning higher scores to the positive clips. Saquil et al. (2021) utilizes multiple pairwise rankers to capture both the local and global information. Badamdorj et al. (2021) assigns higher scores to annotated clips based on dual-modals, *i.e.*, the visual and audio streams. Based on ranking methods, a lot of works aim to mitigate the expensive cost of human annotation using unsupervised techniques or priors, such as (Xiong et al., 2019; Badamdorj et al., 2022). Different from the above methods, regression-based methods predict the locations of highlights directly. Zhu et al. (2021) presents anchor-based and anchor-free approaches to predict the start and end timestamps, as well as the confidence score of highlights, therefore avoiding the laborious manual-designed postprocessing. Moment-DETR (Lei et al., 2021) employs a transformer decoder to obtain the timestamps of specific clips based on different queries. Although the existing methods mentioned above boost the performance of VHD tasks, they substantially neglect the requirements of incremental learning in VHD, which is critical to the practical applications of VHD. In reality, numerous videos and new interests are created rapidly, thus demanding the VHD model to be capable of efficiently handling increasing highlight domains and data.

**Incremental Learning** (IL) is of great concern since the natural vision systems are inherently incremental. The main problem to solve in incremental learning is catastrophic forgetting, manifested as the forgetting of old classes or domains when learning new concepts. As investigated in Lange et al. (2022), the primary efforts deal with this issue in three aspects: using a memory buffer to store the representative data (Rebuffi et al., 2017; Isele & Cosgun, 2018; Rolnick et al., 2019; Yan et al., 2021; Lange & Tuytelaars, 2021), adopting regularization terms to constrain the change of model's weights or outputted logits (Kirkpatrick et al., 2016; Zenke et al., 2017; Schwarz et al., 2018) and performing parameter isolation to dedicate different model parameters for each task (Fernando et al., 2017; Mallya & Lazebnik, 2018; Rosenfeld & Tsotsos, 2020). The memory buffer replays previous samples while learning new concepts to eliminate forgetting, however, it may lead to overfitting on the stored sub-set and incur heavy memory costs. The regularization-based methods penalize the model if some characteristics are changed during the next training stage, resulting in the domination of the so-called essential characters. Parameter isolation grows new branches for new tasks, raising prohibitive colossal architectures. To mitigate the negative effects of existing methods, GPE employs prototype learning combined with distance measurement to perform binary classification (highlight vs. vanilla frames). Prototypes are essentially the most representative features learned across the whole training data, circumventing both the issue of overfitting on the sub-set and the unbearable costs of storing raw data. Besides, we constrain the change of prototypes between stages which is an overall refinement instead of minority domination. The prototypes from previous stages are inherited during training in the current domains, so as to maintain global consistency while leaving room for adjustment and improvements.

## 3 PROBLEM STATEMENT

In incremental VHD, the training procedure consists of several consequent tasks built on disjoint datasets with distribution shifts. Assuming that we have $T$ tasks in total, and this yields a training data stream $\{\mathcal{T}_1, \mathcal{T}_2, ..., \mathcal{T}_T\}$ where $\mathcal{T}_i \cap \mathcal{T}_j = \varnothing$ if $i \neq j$. Each training task $\mathcal{T}_t$ is represented as $\{(x_i^t, y_i^t)\}_{i=1}^{n_t}$ where $x_i^t \in \mathcal{X}$ denotes the whole frame set of *i-th* training video in stage $t$, $y_i^t$ is its corresponding frame-wise label (*i.e.*, a binary vector indicating highlight/vanilla frames), and $n_t$ rep-

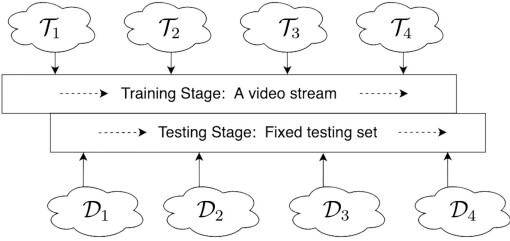

Figure 2: Illustration of conducting incremental VHD on *LiveFood*.

resents the number of accessible training data pairs. Moreover, we use $\{\mathcal{D}_1, \mathcal{D}_2, ..., \mathcal{D}_T\}$ to describe the corresponding domains included in training tasks $\{\mathcal{T}_1, \mathcal{T}_2, ..., \mathcal{T}_T\}$. Note $\mathcal{D}_i \neq \mathcal{D}_j$ if $i \neq j$. Specifically, in our proposed $LiveFood$, we split the training videos into four disjoint subset $\{\mathcal{T}_1, \mathcal{T}_2, \mathcal{T}_3, \mathcal{T}_4\}$, and the corresponding domains are denoted as $\{\mathcal{D}_1, \mathcal{D}_2, \mathcal{D}_3, \mathcal{D}_4\}$. Considering that a video may consist of more than one domain, we further constrain that the domains appearing in $\mathcal{D}_{t-1}$ is a sub-collection of that in $\mathcal{D}_t$. Formally, let $\mathcal{S}_t$ denotes all possible combinations of domains presented in $\mathcal{D}_t$, and $\mathcal{C}_t$ is the domain combinations appearing in videos of $\mathcal{T}_t$, we have

$\mathcal{C}_1 = \mathcal{S}_1$ and $\mathcal{C}_t = \mathcal{S}_t \setminus \bigcup_{i=1}^{t-1} \mathcal{S}_i$. More concretely, let $d_i$ denotes a specific domain label, then if $\mathcal{D}_1 = \{d_1\}$, $\mathcal{D}_2 = \{d_1, d_2\}$, and $\mathcal{D}_3 = \{d_1, d_2, d_3\}$, we have $\mathcal{C}_1 = \{d_1\}$, $\mathcal{C}_2 = \{d_2, (d_1, d_2)\}$ and $\mathcal{C}_3 = \{d_3, (d_1, d_3), (d_2, d_3), (d_1, d_2, d_3)\}$. In above example, $\mathcal{C}_2 = \{d_2, (d_1, d_2)\}$ indicates that the videos in $\mathcal{T}_2$ can contain the domain of $d_2$ or the mixture of $d_1$ and $d_2$. Videos that merely contain the domain of $d_1$ are excluded from $\mathcal{T}_2$ since the intention of incremental VHD is to effectively learn new concepts while remembering what are already learned in past data. The testing set contains mixed videos including **all domains**, and during task $\mathcal{T}_t$, only the domain **appearing in** $\mathcal{D}_t$ is treated as positive when evaluating the performance.

## 4 THE LIVEFOOD DATASET

**Video Selection.** We collect online gourmet videos with high resolution. As introduced in Xiong et al. (2019), shorter videos are more likely to contain attractive clips and thus have more hits, while longer videos are usually boring. Taking this prior into consideration, we filter out both the extremely short (less than 30 seconds) which may contain insufficient gourmet content to learn from and long (over 15 minutes) videos which users generally pay less attention to. After that, all reserved raw videos are viewed by qualified workers to check whether the content of videos is gourmet-related or not, eliminating the effects of incorrect category annotation. Only videos that pass the aforementioned checks are selected for subsequent annotation tasks, in order to guarantee the quality of *LiveFood*. Figure 3 (a) shows the duration distribution of the videos across all domains.

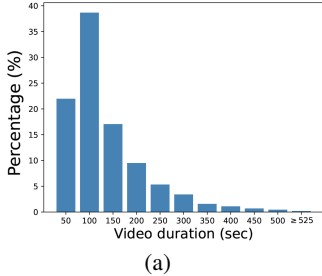 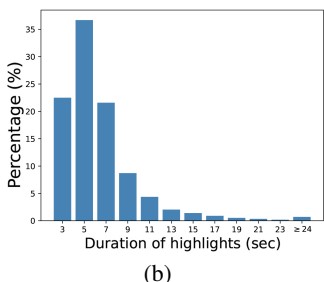 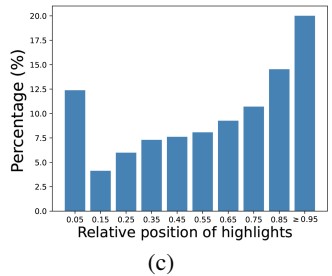

(a)             (b)             (c)

Figure 3: Statistical results of the proposed *LiveFood*. (a) shows the distribution of video duration. (b) illustrates the distribution of highlight durations. Most highlight clips are shorter than 15 seconds. (c) shows the relative position of each attractive clip w.r.t. corresponding videos. The highlights distribute evenly across whole videos, evidencing the good diversity of *LiveFood*.

**Highlights Annotation.** We define four highlight domains that are generally presented in collected videos, namely *cooking*, *eating*, *ingredients* and *presentation*. For each domain, a video clip is accepted as a satisfactory highlight if it meets the following criteria in Table 1.

Table 1: Basic description of annotated domains in *LiveFood*.

| Domain | Description |
|---|---|
| *cooking* | The process demonstrates the chef's exquisite cooking skills. |
| *eating* | People enjoy foods with exaggerated or satisfied expressions on their faces. |
| *ingredients* | The video clip shows at least three kinds of rare ingredients in high quality. |
| *presentation* | A well-displayed meal to make it looks more appetizing. |

The annotators are required to glance over the whole video first to locate the coarse position of attractive clips. Afterward, the video is annotated at frame level from the candidate position to verify the exact start and end timestamps of highlights. Meanwhile, we introduce a strict double-check mechanism (cf. Appendix A.3) to further guarantee the quality of annotations. Since selecting the timestamps of highlights is partly subjective, both objective and subjective verifications are necessary in quality control. Concerning the consistent visual feeling of videos, we restrict the highlights to be longer than three seconds, and less than two minutes to avoid being tedious.

**Data Statistics.** Figure 3 depicts the statistical results of our proposed *LiveFood*, including the distribution of video duration, the distribution of highlights duration, and the relative position of

center timestamp with respect to each video, etc. Besides, in Table 2, we also compare the proposed *LiveFood* with existing VHD datasets such as SumMe (Gygli et al., 2014), YouTubeHighlights (Sun et al., 2014), Video2GIF (Gygli et al., 2016), PHD (Garcia del Molino & Gygli, 2018) and QVHighlights (Lei et al., 2021) for better illustrating the differences among them.

Table 2: Comparison between the proposed *LiveFood* and existing datasets.

| Dataset | Year | Contents | Label domain / class | Total number of videos / highlights | Avg. len. (sec) of videos / highlights |
|---|---|---|---|---|---|
| SumMe | 2014 | Open | ✗ | 25 / 390 | 120.0 / - |
| YouTubeHighlights | 2014 | Activity | ✗ | 712 / - | 143.0 / 2.0 |
| Video2GIF | 2016 | Open | ✗ | 80K / 98K | 332.0 / 5.2 |
| PHD | 2018 | Open | ✗ | 119K / 228K | 440.2 / 5.1 |
| QVHighlights | 2021 | Vlog / News | ✗ | 10.2K / 10.3K | 150 / 24.6 |
| *LiveFood* (ours) | 2022 | Gourmet | ✓ | 5.1K / 14.3K | 136.5 / 6.4 |

As demonstrated in Table 2, the SumMe and YouTubeHighlights only contain a small number of videos and annotations, which makes them insufficient for training deep models. The Video2GIF and PHD are edited by online users and lack strict quality control mechanisms. Thus the reliability of datasets may be undermined. The newly released QVHighlights can not be used for incremental learning since it does not have domain annotations. Besides, the average length of clips in QVHighlights is pretty long: nearly one-fifth of each video is annotated as attractive clips, thus causing the selected clips less discriminative compared to the vanilla clips. Different from the above datasets, *LiveFood* provides gourmet videos with finely annotated domain labels, making it suitable for domain-incremental VHD tasks.

## 5 METHOD

GPE aims to tackle forgetting while still improving by learning new concepts. As analyzed in Section 2, conventional incremental learning methods have shortcomings, such as overfitting on replayed data, limited flexibility, and unbearable growing architectures. Distinguished from them, GPE employs prototypes together with distance measurement to solve the classification problem. Prototypes are compact and concentrated features learned on the training data, mitigating the effects of overfitting on the stored sub-set. In addition, by using global and dynamic prototypes, we endow the model with appealing capability for further refinement when fed with new data ($M_{\mathrm{f}}$ in Figure 4) or accommodation to new domain concepts ($M_{\mathrm{d}}$ in Figure 4).

**Architecture.** Inspired by Carion et al. (2020), GPE employs the combination of convolution- and attention-based models to extract features. Concretely, a ConvNeXt (Liu et al., 2022) pre-trained on ImageNet (Russakovsky et al., 2015) is used to extract spatial features of input video frames. After that, a transformer encoder with multi-heads is used to perform temporal fusion, generating global representations based on the whole video frames. With the transformation of a feedforward network (FFN) which consists of fully-connected layers, each frame is classified based on the distance to learnable prototypes. We aim to learn two groups of trainable prototypes with the same shape, namely the highlight (positive) and vanilla (negative) prototypes. By denoting the dimensionality of output feature of transformer as $m$ and the number of prototypes within each group as $k$, both the highlight and vanilla prototypes can be represented as a matrix with shape $k \times m$. By utilizing $L_2$ distance as the distance measurement between each feature and prototype, we obtain the pair-wise distance between features and each group of prototypes. Formally, we use $h$, $H$, and $V$ to represent the transformer feature, the highlight prototypes and the vanilla prototypes. $g_\phi(\cdot)$ denotes the FFN module. $d(\cdot)$ is the $L_2$ distance between a feature-prototype pair. The distance from feature $h$ to $H$ and $V$ are formulated as:

$$d_H = \min_{i=1:k} d(g_\phi(h), H_i), \quad d_V = \min_{i=1:k} d(g_\phi(h), V_i), \tag{1}$$

where the subscript $i$ represents the $i$-th prototype. The distance is mapped to probability $P_H$ using the softmax function which can be understood as the confidence of assigning feature $h$ to highlights.

$$P_H = \frac{\exp(-d_H)}{\exp(-d_H) + \exp(-d_V)}. \tag{2}$$

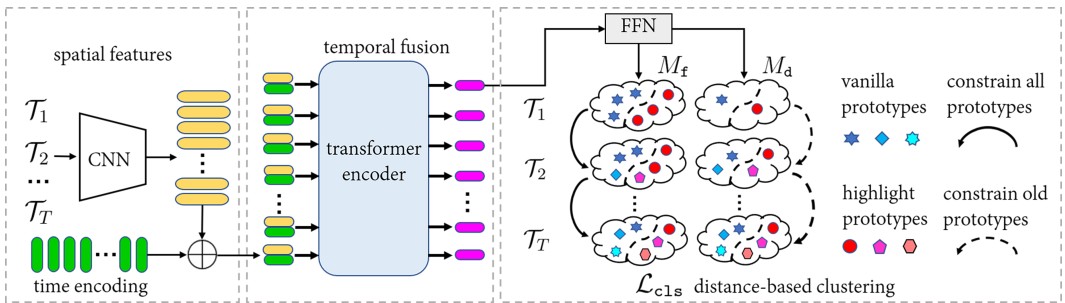

Figure 4: The proposed GPE framework. $\{\mathcal{T}_i\}_{i=1}^T$ indicates a training stream with $T$ tasks. In the right-most part, $M_\mathrm{f}$ and $M_\mathrm{d}$ represents the fixed and dynamic modes of GPE. $M_\mathrm{f}$ defines the number of prototypes in advance and refines them across different stages. A restriction on the magnitude of change amplitude is imposed during learning (cf. Eq. 5). $M_\mathrm{d}$ dynamically adds new prototypes into the learning process when dealing with new domains. The change restriction is only applied on inherited prototypes. This mode is more suitable when learning on a large amount of domains. Each prototype in above figure is equivalent to a row of $V$ or $H$ (cf. Eq. 1).

Then, we use cross-entropy loss to optimize the model through gradient back-propagation:

$$\mathcal{L}_\mathrm{cls} = -\frac{1}{N} \sum_{i=1}^N y_i \cdot \log(P_H) + (1 - y_i) \cdot \log(1 - P_H) \tag{3}$$

where $N$ represents the size of training frames and $y_i$ equals 1 if the $i$-th frame is annotated as highlights otherwise 0.

**Learning with incremental domains.** We detail the learning of $M_\mathrm{f}$ (Figure 4) in this section, and the dynamic mode $M_\mathrm{d}$ can be easily derived by only restricting the change of inherited prototypes and training newly added prototypes freely. We use $h_\theta(\cdot)$ parameterized by $\theta$ to indicate the feature extractor jointly constructed by a ConvNeXt and a transformer encoder. The FFN $g_\phi(\cdot)$ is parameterized by $\phi$. Recall the outputted feature is $h$. Both the vanilla and highlight prototypes are denoted by $\pi$ for simplicity. With the help of these notations, the classification loss built in Eq. 3 is abbreviated as $\mathcal{L}_\mathrm{cls}(\theta, \phi, \pi)$. We expand the definition of distance measurement for evaluating the distance between given two prototypes. For two sets of learned prototypes $\pi^{(t)}$ and $\pi^{(t+1)}$, the distance between them is calculated as:

$$d(\pi^{(t)}, \pi^{(t+1)}) = \frac{1}{k} \sum_{i=1}^k \sqrt{\sum_{j=1}^m (\pi_{i,j}^{(t)} - \pi_{i,j}^{(t+1)})^2} \tag{4}$$

During the training phase $T$, the model inherits trained prototypes $\pi^{(T-1)}$ from the former stage. For the incremental need, we tackle the catastrophic forgetting issue by restricting the change of prototypes, guaranteeing the awareness that the model has learned towards the observed domains. With the above formulations, we consider the following constrained nonlinear optimization problems:

$$\begin{aligned} \min_{\theta,\phi,\pi} \quad & \mathcal{L}_\mathrm{cls}(\theta, \phi, \pi) \\ \mathtt{s.t.} \quad & d(\pi^{(T-1)}, \pi) \leq \gamma \end{aligned} \tag{5}$$

where $\gamma$ is the tolerable change introduced to the observed prototypes. The optimal result that meets the above restriction is $(\theta^{(T)}, \phi^{(T)}, \pi^{(T)})$. Instead of solving the complex nonlinear problem, we resort to its corresponding empirical dual formulation. With an auxiliary positive Lagrange multiplier $\lambda$, the optimization objective in Eq. 5 is transformed into the following manner:

$$\begin{aligned} S^{(T)} &= (\theta^{(T)}, \phi^{(T)}, \pi^{(T)}) \\ &= \max_\lambda \min_{\theta,\phi,\pi} \ L(\theta, \phi, \pi, \lambda) \\ &= \max_\lambda \min_{\theta,\phi,\pi} \ \mathcal{L}_\mathrm{cls}(\theta, \phi, \pi) + \lambda[d(\pi^{(T-1)}, \pi) - \gamma] \end{aligned} \tag{6}$$

where $S^{(T)}$ indicates the optimal solution in training stage $T$. We update the trainable parameters (*i.e.*, $\theta$, $\phi$, and $\pi$) and the empirical Lagrangian variable $\lambda$ alternatively and iteratively:

$$
\begin{aligned}
\theta &\leftarrow \theta - \eta \frac{\partial \mathcal{L}_{\mathtt{cls}}(\theta, \phi, \pi)}{\partial \theta} \\
\phi &\leftarrow \phi - \eta \frac{\partial \mathcal{L}_{\mathtt{cls}}(\theta, \phi, \pi)}{\partial \phi} \\
\pi &\leftarrow \pi - \eta \frac{\partial L(\theta, \phi, \pi, \lambda)}{\partial \pi} \\
\lambda &\leftarrow \max\{\lambda + \eta[d(\pi^{(T-1)}, \pi) - \gamma], 0\}
\end{aligned}
\tag{7}
$$

where $\eta$ is the learning rate of trainable parameters and $\lambda$ is the multiplier in the dual step. With the above analysis, the incremental training and inference pipeline is summarized in Algorithm 1.

---

**Algorithm 1:** Global Prototype Encoding for Incremental Video Highlights Detection.

---

**Input:** training data $\mathcal{X}$ with $k$ stages; initial $h_\theta(\cdot)$, $g_\phi(\cdot)$, and $\pi$; initial $\eta$, $\gamma$, and $\lambda$.
**Output:** trained $h_\theta(\cdot)$, $g_\phi()\cdot$, and $\pi$ at each stage.

| | | | |
|---|---|---|---|
| 1 | **while** *Training* **do** | 10 | **while** *Inference* **do** |
| 2 |   **for** *stage i=1:k* **do** | 11 |   **for** *stage i=1:k* **do** |
| 3 |     Reset $\eta$, $\gamma$, and $\lambda$; | 12 |     **repeat** |
| 4 |     **repeat** | 13 |       Sample $\{x, y\}$ from the testing set at stage $i$; |
| 5 |       Sample a batch data $\{x, y\}$ from $\mathcal{X}$ within stage $i$; | 14 |       Calculate logits: $l = g_\phi(h_\theta(x))$; |
| 6 |       Calculate outputted logits: $g_\phi(h_\theta(x))$; | 15 |       Classify inputs based on the distance between $l$ and $\pi$; |
| 7 |       Calculate the loss function with Eq 6; | 16 |     **until** *the testing set is $\emptyset$*; |
| 8 |       Update $\theta$, $\phi$, $\pi$, and $\lambda$ with Eq 7; | 17 |     Average the mAP cross domains. |
| 9 |     **until** *$\theta$, $\phi$, and $\pi$ are converged*; | 18 |   Average the mAP cross all stages. |

---

## 6 EXPERIMENT

We introduce the details of the evaluation protocol and experimental results in this section.

### 6.1 EXPERIMENTAL SETUP

**Data and Evaluation Protocol.** *LiveFood* contains 4928 videos for training and 261 videos for testing. We randomly split 15% of the 4928 videos for validation. $\mathcal{T}_1$, $\mathcal{T}_2$, $\mathcal{T}_3$ and $\mathcal{T}_4$ consist of 3380, 854, 393, and 113 videos respectively. $\mathcal{D}_1$, $\mathcal{D}_2$, $\mathcal{D}_3$ and $\mathcal{D}_4$ are {*presentation*}, {*presentation, eating*}, {*presentation, eating, ingredients*}, and {*presentation, eating, ingredients, cooking*}. We report the mAP on testing set following previous works (Yao et al., 2016; Xiong et al., 2019).

**GPE.** GPE only finetunes the last layer of ConvNeXt during training. The transformer encoder has 8 heads and 3 layers. The feedforward module $g_\phi(\cdot)$ is a multi-layer perception with 3 linear layers activated by ReLU. Both the vanilla and highlight prototypes are formulated as 40 vectors with dimensions of 128. During every stage, the model is trained with 300 epochs, and the learning rate halves every 70 epochs starting with 1e-3. GPE is initialized randomly during the first stage. In later stages, it trains prototypes in $M_{\mathrm{f}}$ by starting from weights learned in the former stage. In $M_{\mathrm{d}}$, all prototypes learned in previous stages are inherited and trained as similar as in $M_{\mathrm{d}}$. In addition to these inherited prototypes, in each stage, new prototypes are randomly initialized and added to $M_{\mathrm{d}}$ so as to enhance the learning of new concepts.

**Regularization-based methods.** SI (Zenke et al., 2017) and oEWC (Schwarz et al., 2018) are representatives. These schemes update the model with a compromised loss function, utilizing both the overall importance of previous stages and the current task to eliminate forgetting.

**Replay methods.** ER (Rolnick et al., 2019) and DER (Buzzega et al., 2020) use a memory buffer to store the representative data from the previous task to defend against forgetting. The buffer size is set to 200 if not specified.

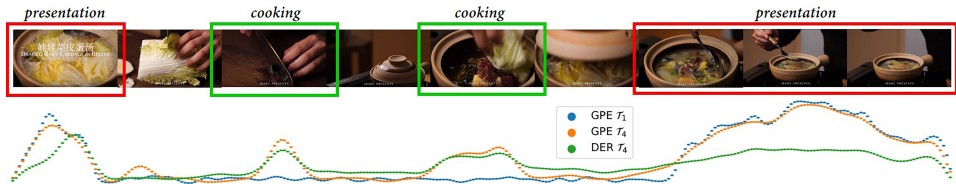

Figure 5: The observed highlights across different training stages.

**Lower bound (Lb).** In each stage, GPE is trained without constraints (cf. Eq. 5), suffering from severe catastrophic forgetting and rendering performance drops with the increasing of tasks.

**Upper bound (Ub).** In each stage, GPE is trained with data from all stages. Thus it is free from the forgetting issue and provides the upper bound performance for all incremental learning methods.

## 6.2 MAIN RESULT

**Comparison with existing IL methods**. The experimental results are depicted in Table 3. We highlight the upper bound results with gray background. It can be observed from Table 3 that the vanilla GPE ($M_{\mathrm{f}}$) surpasses the lower bound with an improvement of 2.91% mAP. Compared to the classic IL methods SI and oEWC, GPE outperforms them by a remarkable margin, yielding at least 1.57% performance gain on mAP. Moreover, when equipped with the same replay schemes as Yan et al. (2021), GPE achieves 1.06% and 2.71% mAP gain compared to DER and ER. The above results clearly demonstrate the effectiveness of GPE in tackling the incremental VHD task.

Table 3: Comparison of GPE with existing incremental learning methods on *LiveFood*. We evaluate their frame-wise mAP performance.

| mAP | Lb | SI | oEWC | ER* | DER* | Ub | GPE($M_{\mathrm{f}}$) | GPE($M_{\mathrm{d}}$) | GPE* |
|---|---|---|---|---|---|---|---|---|---|
| $\mathcal{T}_1$ | 36.13 | 36.16 | 36.13 | 35.79 | 36.17 | 36.15 | 36.14 | 36.21 | 36.17 |
| $\mathcal{T}_2$ | 30.86 | 31.84 | 31.82 | 31.38 | 33.14 | 37.38 | 35.82 | 36.13 | 36.62 |
| $\mathcal{T}_3$ | 29.18 | 30.72 | 30.51 | 29.13 | 32.52 | 36.90 | 31.87 | 32.74 | 33.15 |
| $\mathcal{T}_4$ | 25.89 | 28.73 | 28.67 | 29.06 | 30.11 | 36.30 | 29.88 | 30.15 | 30.27 |
| Avg. | 30.52 | 31.86 | 31.78 | 31.34 | 32.99 | 36.68 | 33.43 | 33.90 | **34.05** |

*: using memory buffer to replay samples.

**Visualization of highlight scores across training stages.** We investigate the effects of observed prototypes across different training stages. In Figure 5, we present the highlight detection results of GPE in the first and the last training stage. For comparison, we also provide the prediction of DER. In the curves shown in Figure 5, the blue and orange points indicate the highlight scores of each frame predicted by GPE (*i.e.*, $P_H$ in Eq. 2) during the first task $\mathcal{T}_1$ and the final task $\mathcal{T}_4$. The green points represent the predicted scores of DER in $\mathcal{T}_4$. It is clear that GPE can learn new concepts, *e.g.*, *cooking* while keeping the memory of *presentation* learned in the first stage. This result is in line with our motivation that a strong incremental VHD model should be able to cover both the past and new concepts. In contrast, since DER employs stored data to strengthen memory, it still has the drawback of being prone to forgetting due to the limited buffer size. This is demonstrated by its assigning much lower scores to the old domain of *presentation* when learning *cooking*.

**Scalability of GPE in the dynamic manner.** We investigate the generalization ability of dynamic GPE ($M_{\mathrm{d}}$) in the scenario where the model needs to handle a large number of domains. We consider the R-MNIST dataset containing a series of rotated digits with different degrees between $[0, \pi)$, where each degree represents a domain. For a fair comparison, we use the identical settings as Buzzega et al. (2020), yielding a stream with 20 subsequent tasks. Note that no augmentation techniques are used. In this experiment, GPE is simplified to be a small network with 2 fully-connected layers followed by ReLU. The number of prototypes is set to 5 per class. Therefore each task $\mathcal{T}_t$ has $5t$ prototypes per class by inheriting from previous stages and generating 5 new prototypes for each class. The old prototypes are forbidden to change too much. $\lambda$ and $\gamma$ are 10 and 1e-2. All other experimental settings follow Buzzega et al. (2020).

Table 4: Average classification accuracy on R-MNIST.

| Method | SI | oEWC | ER | GEM | FDR | GSS | HAL | DER | GPE ($M_d$) | GPE ($M_d$) |
|---|---|---|---|---|---|---|---|---|---|---|
| Buffer Size | ✗ | ✗ | 200 | 200 | 200 | 200 | 200 | 200 | ✗ | 200 |
| Avg Acc. | 71.91 | 77.35 | 85.01 | 80.80 | 85.22 | 79.50 | 84.02 | 90.04 | $85.42_{\pm 0.13}$ | $\mathbf{90.17}_{\pm 0.25}$ |

Results depicted in Table 4 demonstrate that the dynamic GPE surpasses most conventional methods, including the regularization-based and replay methods. Notably, the dynamic GPE achieves 85.42% average accuracy across 20 tasks, comparable to ER (Rolnick et al., 2019) and FDR (Benjamin et al., 2019) while outperforming other methods with a considerable margin. We further adopt the identical replay schemes as DER (Yan et al., 2021), and this helps the dynamic GPE achieve 90.17% top-1 accuracy, which is established as a new comparable *state-of-the-art* approach.

## 6.3 ABLATION STUDY

**Ablation on the initial number of prototypes $k$.** The number of prototypes reflects the model's capacity. Too many prototypes increase the training cost while too few lead to underfitting. Results shown in Table 5 provide a comparison between the training cost and performance under the fixed mode of GPE ($M_f$) on *LiveFood*. It is observed that with the increasing initial quantity of prototypes $k$, the average mAP over all tasks consistently increases. However, by comparing the last two rows in Table 5, we notice that the performance gain between $k = 40$ and $k = 50$ is marginal though more parameters are introduced. Therefore, we set $k$ to 40 throughout experiments to strike a good balance between accuracy and efficiency.

Table 5: Ablations on the initial number of prototypes $k$.

| $k$ | mAP ($\gamma = 5.0$) | | | | |
|---|---|---|---|---|---|
| | $\mathcal{T}_1$ | $\mathcal{T}_2$ | $\mathcal{T}_3$ | $\mathcal{T}_4$ | Avg. |
| 10 | 35.31 | 33.48 | 29.18 | 25.98 | 30.99 |
| 20 | 35.82 | 34.66 | 30.72 | 28.73 | 32.35 |
| 30 | 36.13 | 35.70 | 30.51 | 28.67 | 32.75 |
| 40 | 36.14 | 35.82 | 31.87 | 29.88 | 33.43 |
| 50 | 36.21 | 35.88 | 31.90 | 29.94 | **33.48** |

Table 6: Ablations on the changing constraints of distance $\gamma$.

| $\gamma$ | mAP ($k = 40$) | | | | |
|---|---|---|---|---|---|
| | $\mathcal{T}_1$ | $\mathcal{T}_2$ | $\mathcal{T}_3$ | $\mathcal{T}_4$ | Avg. |
| 1e-3 | 36.14 | 34.26 | 30.17 | 26.43 | 31.75 |
| 1.0 | 36.13 | 34.97 | 30.62 | 27.41 | 32.28 |
| 3.0 | 36.15 | 35.50 | 31.44 | 28.27 | 32.84 |
| 5.0 | 36.14 | 35.82 | 31.87 | 29.88 | **33.43** |
| 15.0 | 36.14 | 34.92 | 30.80 | 28.12 | 32.50 |

**Ablation on distance constraint $\gamma$.** Extremely small $\gamma$ hinders the model from learning new concepts since the prototypes are almost unchanged. In contrast, too large $\gamma$ may lead to catastrophic forgetting since the model may heavily overfit to the newly observed data. As shown in Table 6, we set $k$ to 40 by default and investigate the effects of $\gamma$ with different values. In our experiments, we find the distance between the vanilla and highlight prototypes after $\mathcal{T}_1$ is less than 15, so it is set as the upper bound of $\gamma$. From Table 6, we can see that when $\gamma$ is small, say 1e-3, GPE can hardly learn the new contents, only achieving similar mAP compared to the lower bound method as shown in Table 3. By enlarging $\gamma$, the average mAP increases consistently from 31.75 to 33.43. When $\gamma$ is 15, the model suffers from the forgetting issue as explained before, resulting in a near 1.0% performance drop. Consequently, we set $k$ and $\gamma$ to 40 and 5 by default.

## 7 CONCLUSION

In this paper, we introduce a new task: incremental video highlight detection, aiming to perform VHD in the practical scenario where both the highlight domains and data increase over time. To pave the road in this new direction, we collect a high-quality video gourmet dataset *LiveFood* which contains four fine-annotated domains. Then we propose a new end-to-end model named **G**lobal **P**rototype **E**ncoding (GPE) to learn incrementally to adapt to new highlight domains. Extensive experimental results clearly demonstrate the effectiveness of our method. We hope this work serves to inspire other researchers to work on this new and critical task.

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

# A APPENDIX

## A.1 ETHICS, DATA, AND PRIVACY

We solemnly claim that we strictly hold to the highest standard to obey any regulations/rules regarding ethics codes, data safety, and privacy preservation. Below we introduce the main measures taken to eliminate potential risks/issues.

**Data Desensitization.** All videos appearing in *LiveFood* are crawled from public video-sharing platforms. We remove all private information related to video owners to preserve privacy, including but not limited to identifiers, meta-data, addresses, and profiles.

**Data Storage.** We emphasize that only video owners have the right to retain or remove their uploaded videos on the platform. To protect the intelligent property of creators, we do not store any videos locally. The *LiveFood* consists of video links and corresponding domain annotations. Videos appearing in *LiveFood* turn to invalid after being deleted by the creators.

**User Privacy.** We highly respect user privacy during the construction of the dataset. All videos in *LiveFood* are publicly available on video-sharing platforms. Consent for public usage of videos, including academic research, has been reached between the platform and users. All sensitive data about users have been removed from our dataset.

**Licence.** *LiveFood* can only be used for non-commercial purposes. Applicants must sign an agreement before using the dataset (cf. Appendix A.7).

**Annotator Related.** The annotators are skilled and experienced in data preparation. They have been sufficiently instructed to beware of how to avoid any potential dangers and risks during annotation. Their work is properly compensated per local law.

## A.2 VISUALIZATION OF VIDEO ANNOTATIONS

Due to space limits, we do not present samples for each video domain in the main text. To help better understand *LiveFood*, we show randomly selected videos video_1/.../7 from *LiveFood* along with domain annotations in Figures 6 to 12. To preserve privacy, all faces are blurred.

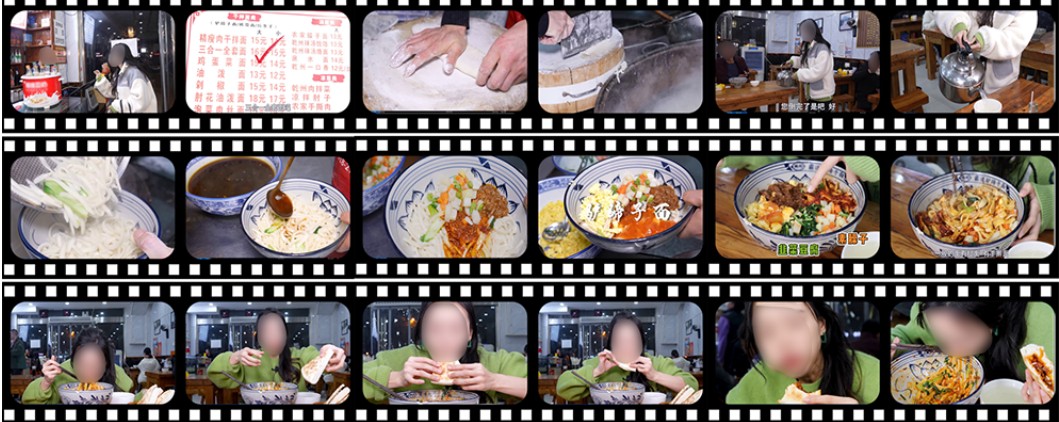

Figure 6: Samples of clips and domain annotations from video_1. The row from top to bottom illustrates vanilla clips, *presentation*, and *eating*, respectively. The faces appearing in video above are blurred to protect privacy.

## A.3 ANNOTATION QUALITY CONTROL

We have applied elaborate quality control to guarantee the high quality of *LiveFood*. Specifically, to reduce subjectivity during annotation, an annotated video is reviewed by another experienced reviewer to double-check if it meets the requirements in Table 1. An annotated video only passes double-checking if both the annotator and reviewer agree on the correctness of domain labels over 90% video segments. Besides, reviewers are also required to identify the start and end timestamps of

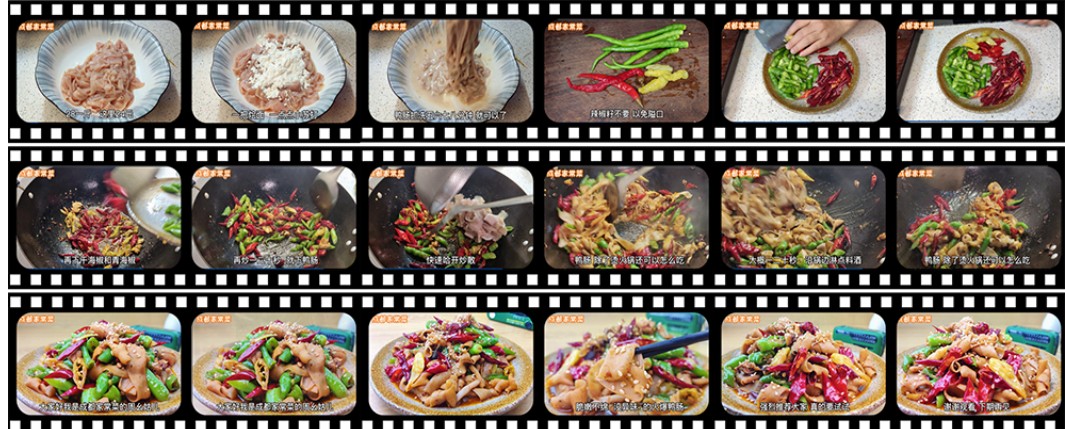

Figure 7: Samples of clips and domain annotations from video_2. The row from top to bottom illustrates examples of *ingredients*, *cooking*, and *presentation*, respectively.

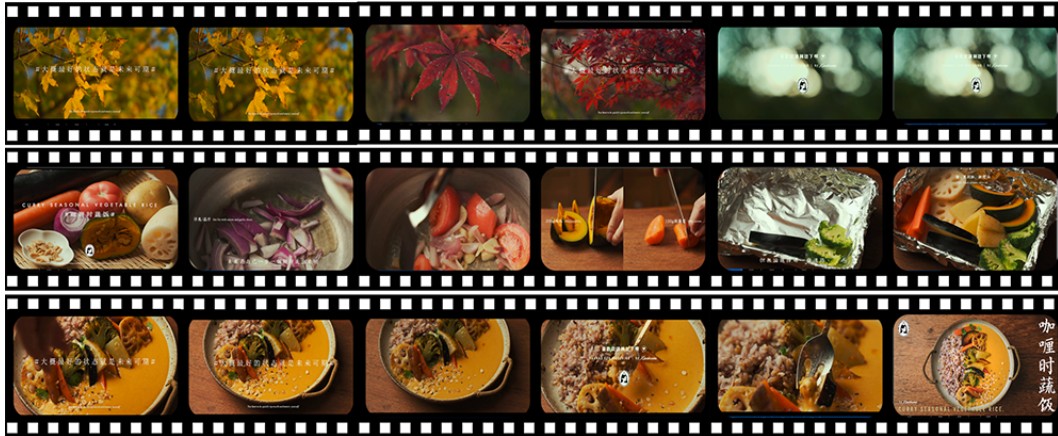

Figure 8: Samples of clips and domain annotations from video_3. The row from top to bottom illustrates examples of vanilla clips, *ingredients*, and *presentation*, respectively.

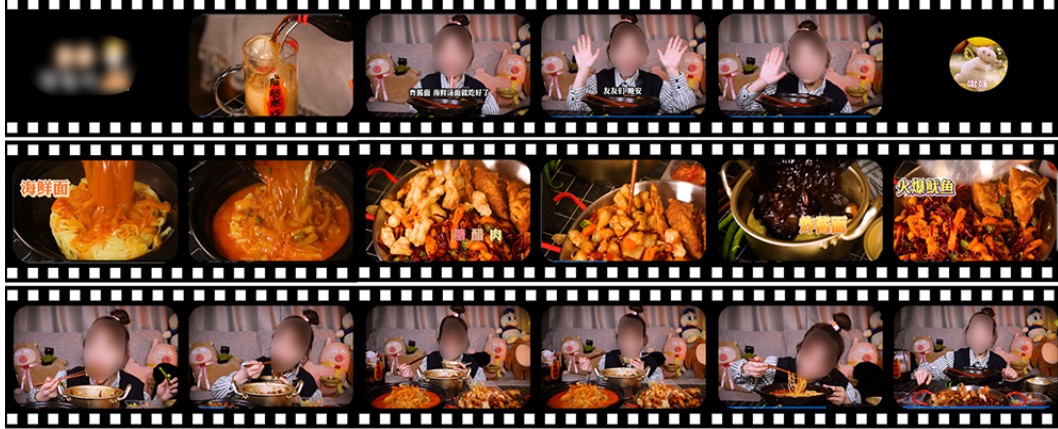

Figure 9: Samples of clips and domain annotations from video_4. The row from top to bottom illustrates examples of vanilla clips, *presentation*, and *eating*, respectively. The faces appearing in video above are blurred to protect privacy.

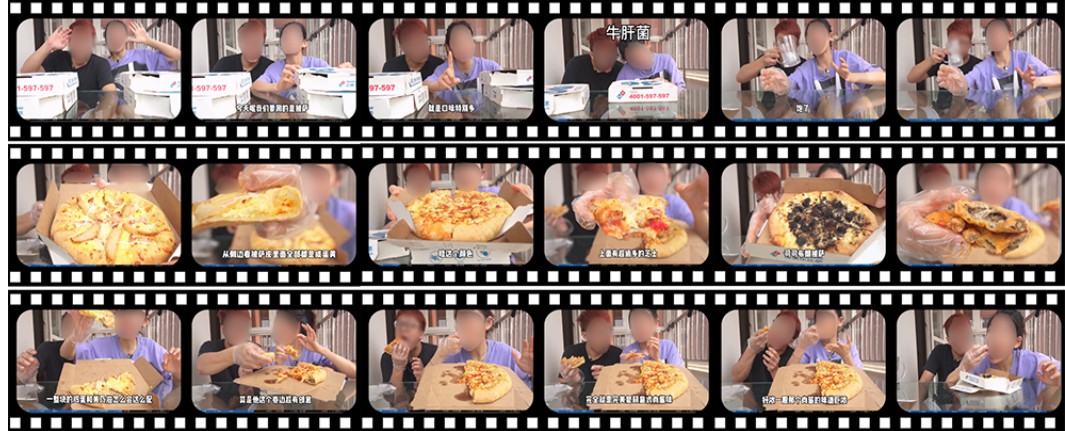

Figure 10: Samples of clips and domain annotations from video_5. The row from top to bottom illustrates examples of vanilla clips, *presentation*, and *eating*, respectively. The faces appearing in video above are blurred to protect privacy.

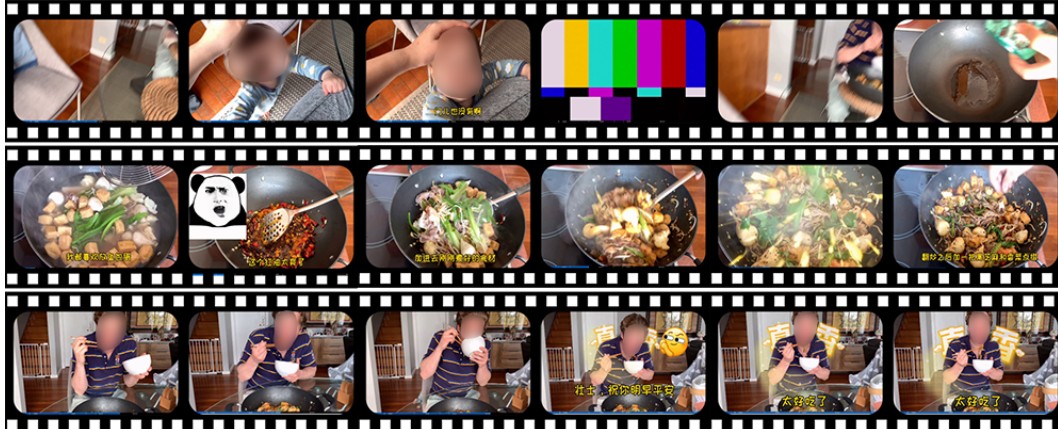

Figure 11: Samples of clips and domain annotations from video_6. The row from top to bottom illustrates examples of vanilla clips, *cooking*, and *eating*, respectively. The faces appearing in video above are blurred to protect privacy.

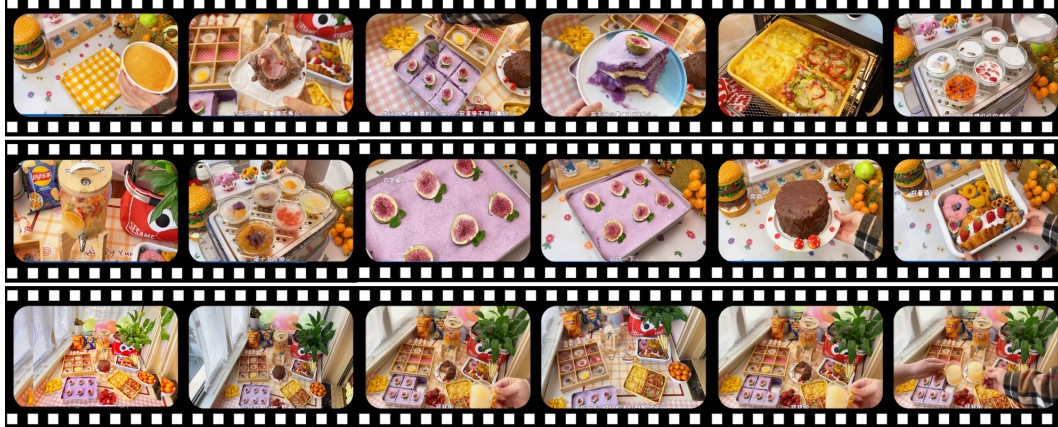

Figure 12: Samples of clips and domain annotations from video_7. The row from top to bottom all illustrates the examples of *presentation*.

highlight clips, and only clips with over 90% temporal IoU (Intersection-over-Union) between the annotations of the annotator and reviewer are deemed as qualified highlight clips. The review process is conducted in a batch-wise manner: 30 of every 100 annotated videos are selected randomly for inspection. The threshold of qualification rate is 90%. Before starting the official annotation, all annotators are trained with over one hundred testing videos to get fully prepared.

## A.4 IN-HOUSE ANNOTATION TOOL

We use an in-house tool named *LiveLabel* for annotating videos. In this section, we introduce the basic functions of *LiveLabel* to give a rough idea of how it is used in the annotation task.

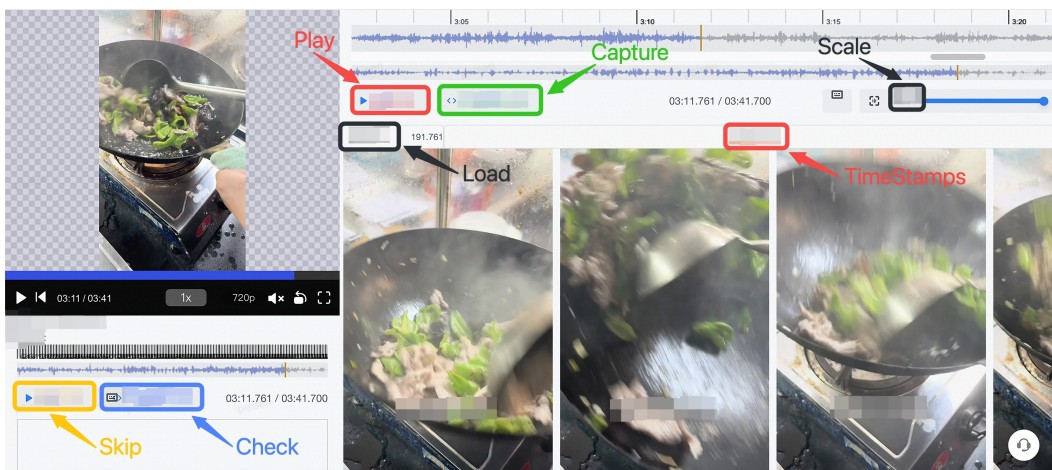

Figure 13: The annotation tools used in *LiveFood*.

As shown in the interface above, *LiveLabel* pulls videos from a pre-defined video pipeline automatically and parses these videos at an adjustable frame rate. On the left panel of Figure 13, the raw video is played at the user-specified speed. One can play/stop the video display using the *play* button. Pushing *capture* will demonstrate the detailed video contents in a frame-by-frame fashion, as shown in the right panel of Figure 13. These frames help the annotators select accurate timestamps for video highlight clips. The *check* button is used to select pre-defined domains or categories. If a video is undesired, the annotator can click the *skip* button to remove them from *LiveFood*.

## A.5 MORE COMPARISON RESULTS ON R-MNIST

In Table 4, we show the comparison results between GPE and popular incremental learning approaches on the R-MNIST benchmark. We note that all methods employ identical architecture for a fair comparison, *i.e.*, a multi-layer perception (MLP) with two fully connected layers. In this section, we further investigate the extensibility of GPE to stronger backbones, *e.g.*, the convolution network as in (Cha et al., 2021). Since detailed structural information of the convolution network used in Cha et al. (2021) is unavailable, we use ResNet-18 (He et al., 2016) as the backbone for GPE and all other competing methods. For a comprehensive comparison, we experiment with the following three settings: (1) using no memory buffer, (2) using a memory buffer of 200 samples, 3) using a memory buffer of 500 samples. We directly take the results from the paper of $Co^2L$ (Cha et al., 2021) for comparison. For references of other methods in Table 7, please refer to: ER (Rolnick et al., 2019), GEM (Lopez-Paz & Ranzato, 2017), A-GEM (Chaudhry et al., 2019), FDR (Benjamin et al., 2019), GSS (Aljundi et al., 2019), HAL (Chaudhry et al., 2021), DER (Yan et al., 2021), DER++ (Yan et al., 2021), and $Co^2L$ (Cha et al., 2021).

When **no** memory buffer is applied, GPE with ResNet-18 (He et al., 2016) backbone achieves **94.77%** top-1 classification accuracy on the R-MNIST dataset. By using a memory buffer of 200 samples and 500 samples, GPE boosts its performance by 2.29% and 3.56%, respectively. Compared to more complex incremental learning techniques such as $Co^2L$ (Cha et al., 2021), although GPE achieves comparable results, it is much simpler and more efficient. For example, $Co^2L$ counteracts

Table 7: Average classification accuracy on R-MNIST using convolution-based models.

| Buffer | ER | GEM | A-GEM | FDR | GSS | HAL | DER | DER++ | Co$^2$L | GPE ($M_{\mathrm{d}}$) |
|---|---|---|---|---|---|---|---|---|---|---|
| 200 | 93.53 | 89.86 | 89.03 | 93.71 | 87.10 | 89.40 | 96.43 | 95.98 | **97.90** | 97.06 |
| 500 | 94.89 | 92.55 | 89.04 | 95.48 | 89.38 | 92.35 | 97.57 | 97.54 | **98.65** | 98.33 |

knowledge forgetting using strong augmentations and distillation penalty, while GPE is much simpler and cleaner by discarding all these tricks. This appealing property makes GPE more scalable to handle practical scenarios. In comparison, Co$^2$L has to design specific augmentation schemes as well as hyper-parameters appearing in the distillation procedure, all leading to the hardship of scaling to large data/tasks.

### A.6 VISUALIZATION OF LEARNED PROTOTYPES

To provide a better understanding of the learned vanilla/highlight prototypes in GPE, we use t-SNE (Van der Maaten & Hinton, 2008) to illustrate their distributions. GPE in the dynamic manner is used in this experiment. Concretely, we conduct two experiments by setting the number of vanilla/highlight prototypes to 40 and 80 respectively. We use scatters in different colors to indicate the various training tasks. $\mathcal{T}_1$, $\mathcal{T}_2$, $\mathcal{T}_3$, and $\mathcal{T}_4$ are used to indicate the prototypes learned at different stages. Figure 14 depicts the learning procedure.

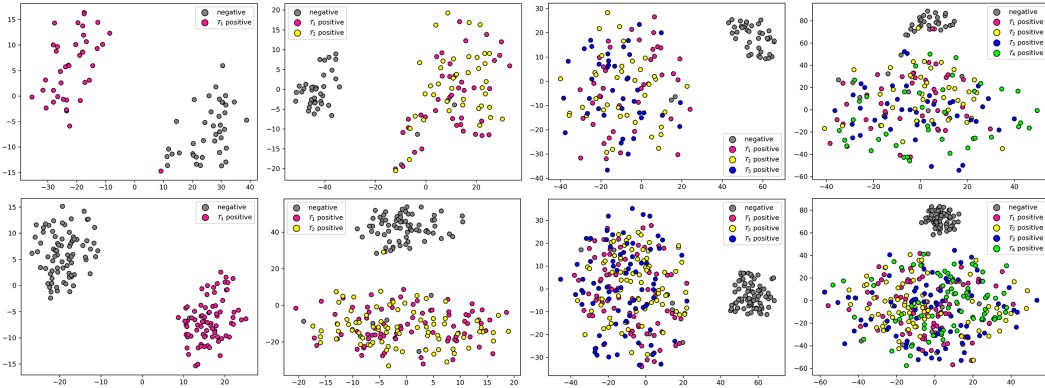

Figure 14: The learned prototypes of GPE across different training stages.

In Figure 14, the number of learnable prototypes is set to 40 in the top row and 80 in the bottom row. Taking the first two images in the top row as examples, GPE is trained to classify the vanilla and *presentation* clips as illustrated in the first image. The scatters in gray are the negative prototypes, while the pink ones are positive (highlight) prototypes of *presentation*. During the second task, GPE aims to learn the characters of *eating* while still retaining that of *presentation*. Therefore, GPE limits the change of prototypes observed in $\mathcal{T}_1$ (in pink) and develops a new group of prototypes in yellow, namely $\mathcal{T}_2$ positive as shown in Figure 14. We emphasize that positive prototypes for $\mathcal{T}_2$ are updated without constraints while that in pink (*i.e.*, $\mathcal{T}_1$ positive) are forbidden to change too much (cf. Eq. 5). The following tasks are optimized similarly.

Based on the depicted images, we can find that GPE adaptively adds new positive prototypes within incremental tasks and meanwhile modifies the position of negative prototypes which have fixed quantity. Since we model the task in each stage as a binary classification problem (highlight vs. vanilla), the goal of GPE is thus to reduce the discrepancy across old and new domains while boosting the distinction between the vanilla and highlight prototypes. The visualization results shown in Figure 14 are consistent with our expectation by showing smaller inter-domain discrepancy and greater distances between positive/negative prototypes. From another view, since we allow the learned prototypes to change for adapting to the new data, both the inherited and the newly developed prototypes are updated simultaneously, thus helping reduce the discrepancy of prototypes learned across domains.

## A.7 *LiveFood* RELEASE AGREEMENT

One must sign the following agreement in order to be permitted to use the *LiveFood* dataset.

---

### *LiveFood* Release Agreement

- The released *LiveFood* can be only used for non-commercial purposes, including academic research and education. Applicants are forbidden to use this dataset for profitability or infringement activities, including but not limited to advertising, selling, face-based applications, *etc*.

- Applicants promise that they will not conduct any form of analysis with respect to bio-info in this dataset, including but not limited to **faces**, **gender**, *etc*.

- Applicants are aware that they take full responsibility for their usage of *LiveFood*. They agree that we reserve the right to ask them to immediately stop illegal behavior and eliminate negative influences caused.

- Applicants agree that we reserve the right to stop their usage of the dataset and delete its copies in found of any violation of this agreement.

- Applicants must clearly state their purposes and the potential effects of using *LiveFood*.

**Name**: ____ **Email**: ____ **Organization**: ____ **Date**: ____ **Signature**: ____

---

