# OpenReview forum: "Global Prototype Encoding for Incremental Video Highlights Detection"
_ICLR.cc/2023/Conference — Submitted to ICLR 2023_

### Official Review · Reviewer_HURe · 2022-10-22

**Confidence:** 3
**Correctness:** 3
**Technical Novelty And Significance:** 3
**Empirical Novelty And Significance:** 3
**Recommendation:** 8

**Clarity, Quality, Novelty And Reproducibility:**

The paper is well written. They provide satisfactory details to understand the concepts, and potential implementation in the manuscript. They also provided the source code, although I wasn't able to run the code due to time constraints.

**Strength And Weaknesses:**

The idea of learnable global prototypes leads to a simpler video highlight detection framework and an apt method for incremental learning. The dataset contribution is also significant, as the previous related datasets are either smaller in scale or do not have domain tags.

**Summary Of The Paper:**

First, the paper introduces a new task of incremental video highlights detection aiming to adapt to more data and new domains. To that end, they propose a framework Global Prototype Encoding (GPE). They employ transformer-based temporal encoder to aggregate frame-level CNN features to generate temporally aware representations of a video. Next, each frame is classified by two groups of learnable prototypes -  highlight prototypes and vanilla prototypes. The utilization of prototypes help incremental learning by keeping the distance between the learned prototype in the new domain and previously observed prototypes close to each other.  A major contribution of this paper is the LiveFood dataset. Performance of their incremental GPE method are compared with other incremental methods relying on complex exemplar selection or complicated replay schemes, and the author observed that GPE produces encouraging results in the incremental setting.

**Summary Of The Review:**

The paper presents a new task - incremental video highlights detection. The framework is based on simple ideas around global prototype encodings. The dataset contribution is also significant.

---

> ### Author Response · Authors · 2022-11-08
> **Response of authors**
>
> We appreciate the reviewer's positive comments on our work. Just as pointed out in the comments, we investigate a novel yet practical task namely **incremental video highlights detection (VHD)**, build a high-quality dataset (**LiveFood**) for facilitating future research and propose **GPE** to address the incremental VHD.  We wish our work inspires more researchers to explore this research field.  Besides, we have added more visualization results and analysis in the Appendix for your reference. Please feel free to raise any other questions or comments.

---

### Official Review · Reviewer_6gVY · 2022-10-23

**Confidence:** 4
**Correctness:** 3
**Technical Novelty And Significance:** 2
**Empirical Novelty And Significance:** 2
**Recommendation:** 3

**Clarity, Quality, Novelty And Reproducibility:**

It is unclear whether the performance increase is merely due to ConvNext/transformer. Although the dataset looks original, the paper lacks detailed information about the dataset. The proposed method does not seem to outperform the work published in ICCV 2021.

**Details Of Ethics Concerns:**

The dataset contains a lot of scenes where people and their faces are present. I believe that it might arouse some ethical concerns so the authors should clarify this.

**Strength And Weaknesses:**

Strength:
- The introduced task of incremental VHD looks interesting and quite practical.
- The proposed dataset has domain annotations that are missing in the previous VHD datasets.
- The presented method achieves state-of-the-art performance on the dataset.

Weakness:
- W1. First of all, the paper does not provide enough information about the proposed dataset, even if it is the major contribution of the paper. The most important and distinctive aspect of the dataset is the presence of domain annotations, but there is no visualization of the four domains. For example, the "eating" domain is not depicted in any of the figures. Moreover, the information on the labeling interface or annotation tool is also missing. I also believe that the quality control section can be more elaborated for better understanding.

- W2. According to the description of annotated domains in Table 1, the "eating" scene contains people and their faces. I believe that it might arouse some ethical concerns so the authors should have stated this in the paper.

- W3. Apart from the dataset, the proposed method does not have a technical novelty. It seems like GPE is merely a combination of ConvNext, transformer encoder, and prototypical learning. For example, all the equations from 1 to 7 do not convey any novel concept because they explain a simple softmax function (eq. 2), BCE loss (eq. 3), L2 distance (eq. 4), etc. I wonder how exactly GPE is different from the previous incremental learning methods based on prototypes. As far as I know, prototypical networks have been used a lot in class-incremental learning, and I also would like to know what novelties GPE has when compared to [ref1] and [ref2]. Especially, [ref1] looks very similar to GPE.

- W4. In addition, GPE's higher performance seems merely due to the use of the improved feature encoders, i.e ConvNext and a transformer. If that is the case, we cannot say that the proposed method using the prototypes contributes to the performance increase because other methods can also benefit from the high-performing feature encoders. At this point, I am not even sure why the authors mentioned "Inspired by Carion et al., GPE employs a ConvNext, ..."; I wonder how DETR inspires the usage of ConvNext.

- W5. The paper lacks an in-depth qualitative analysis of the prototypes. I strongly believe that the authors should visualize the learned prototypes in the feature space to validate the effectiveness of the proposed learning strategy.

- W6. There exists another important baseline called Co2L that was published in ICCV 2021 [ref3]. I don't think GPE is a new state-of-the-art method in terms of the average classification accuracy on R-MNIST, which is wrongly argued by the authors in Table 4.


[ref1] Constrained Few-shot Class-incremental Learning, Hersche et al. (CVPR 2022)

[ref2] Prototype Augmentation and Self-Supervision for Incremental Learning, Zhu et al. (CVPR 2021)

[ref3] Co2L: Contrastive Continual Learning, Cha et al. (ICCV 2021)

**Summary Of The Paper:**

This paper proposes a new task of video highlight detection (VHD) in a domain-incremental setting. First, the authors define the task of incremental video highlights detection and introduce a gourmet dataset named LiveFood that they've carefully collected to facilitate research in this new task. Second, they argue that all the previous approaches fail to address this new task and have limited scalability. To this end, the authors present a new method named Global Prototype Encoding (GPE) that learns two groups of prototypes that are used to optimize a classification loss for incremental learning. The proposed method shows good performance on R-MNIST as well as LiveFood.

**Summary Of The Review:**

The paper has two contributions, which are LiveFood and GPE; but both of them have non-negligible issues.

---

> ### Author Response · Authors · 2022-11-09
> **Response to the proposed concerns.**
>
> We appreciate your valuable comments and suggestions. To address your concerns, we provide all required information in the `Appendix`, including `dataset details`, `visualization`, `more analysis`, and `method comparison`, etc. Please refer to the `Appendix` in the latest manuscript for details. Below are the answers to each question.
> - **Visualization, labeling tool, and quality control (W1)**. Due to space limit, we do not show illustrations of annotations. In `Appendix A.2`, we present a bunch of video samples along with their domain annotations, including "eating", "cooking", "ingredients" and "presentation". For the details of labeling tool and annotation process, please refer to `Appendix A.4`. Please also feel free to check out our response to `Reviewer LRmd` in **Raw data collection** and **Data filtering and annotation** for reference.
> - **Ethics and privacy concerns (W2).** We stress that we hold the highest standard to ethics- and privacy-related regulations in constructing the dataset. We exhaustively introduce measures taken (grouped by **Data Desensitization**, **Data Storage**, **User Privacy**, **Licence**, and **Annotator Related**) to eliminate any potential risks/issues. Please refer to `Appendix A.1` for details. Notably, we only provide the links to videos, the accessibility of which is solely controlled by video owners/creators. Any user of our dataset must sign an agreement (`Appendix A.7`) that strictly confines the use of LiveFood to research purposes only. To preserve privacy, all faces presented in video samples in Appendix are heavily **blurred**.
> - **Novelty (W3).** We note the contributions of our work are three-fold: introduce for the first time a new and practically critical task of **incremental VHD**, release a **high-quality dataset**, and propose a simple yet **strong baseline (GPE)** for tackling this task.  In terms of methodology, GPE is distinguished from other prototype-based incremental learning methods on both the training schemes of prototypes and model architecture. Benefiting from the novel designs, GPE offers a promising solution by striking a good balance between performance and scalability, while other methods may struggle when directly applied to incremental VHD. Below we highlight the novelty and advantages of GPE by comparing GPE with methods ([Ref. 1/2/3]) mentioned by the reviewer. We will also include them in the final version of draft.
>   - **Compare with C-FSCIL (Ref1).** The differences between C-FSCIL (Ref1) and GPE are significant. First, GPE updates previously learned prototypes in training with new data while C-FSCIL uses the fixed mean of class-wise vectors as prototypes; **Hence, GPE is more capable of fitting in the new domain/classes.** Second, GPE does not need memory buffers to store extra information while C-FSCIL depends on an explicit memory buffer (EM) and a global average activation memory buffer (GAAM) to record the state of the model; **Therefore, GPE is more memory/storage efficient.** Third, GPE is end-to-end trainable with simple training settings, while C-FSCIL requires complex meta-learning priors. **Thus, the pipeline of GPE is more scalable and extensible in handling large amounts of videos from multiple domains in practical usage.**
>   - **Compare with PASS (Ref 2).** Different from PASS which directly combines old and new prototypes to deal with forgetting, GPE adaptively optimizes learned prototypes in old domains with constraints, such that both old and new knowledge can be better captured in a single model.
>   - **Compare with Co$^2$L (Ref 3).** Please see our response to **W6** below.

---

> > ### Author Response · Authors · 2022-11-17
> > **Response to the left concerns.**
> >
> > - **Clarification on comparison with other methods (W4).** We believe this question is due to misunderstandings of how we compare different methods. We are sorry for the confusion caused. **We note that all other methods use the same feature extractor as GPE,** i.e., ConvNext and Transformer as GPE, therefore, the performance superiority of GPE is ascribed to nothing but the stronger capability of GPE in learning and remembering representations for incremental video highlight domains. To further clarify this, we explain the experimental results on LiveFood (`Table 3`) and R-MNIST (`Table 4`) in more detail.
> >   - In `Table 3`, we compare GPE with different incremental learning methods, including the regularization-based and reply-based methods. **All of these methods employ the same feature extractor as GPE,** and the only difference is their **training schemes**. For example, both ER and DER use a memory buffer to store videos in previous stages, while SI and oEWC limit the change of the model's parameters. Unlike the above approaches, GPE leaves room for the adaptation of learned prototypes while developing new prototypes to capture novel information. This training manner, on the one hand, **eases the overfitting issue** of the replayed samples; on the other hand, it **strengthens the stability and capability** of the model. Benefiting from the novel designs, GPE outperforms the aforementioned methods with large margins, clearly validating its effectiveness.
> >   - Similarly, in the experiments on R-MNIST (see `Table 4` and `Table 7`), **all comparing methods employ the same backbone,** i.e., a multi-layer perception (`MLP`) with two fully connected layers. As can be observed from `Table 4`, GPE still outperforms others under the fair comparison.
> >   In summary, we convincingly demonstrate that the advantages of GPE come only from its better **training scheme** than others, through comprehensive experiments conducted under fair comparison settings.  We mention "inspired by DETR ..." only for stating the fact that GPE employs a similar architecture consisting of convolution- and attention-based models as in DETR. We are sorry for the confusion caused. This part has been rephrased in the latest manuscript.
> > - **Analysis of learned prototypes (W5).** Both the visualization of learned prototypes and analysis are presented in `Appendix A.6`. Please refer to that section for more details.
> > - **Comparison with Co$^2$L (W6).** By comparing the experimental setups between Co$^2$L (`Experimental Setup (Training)` in Section 5.1 in the paper of Co$^2$L) and GPE (cf. `Scalability of GPE in the dynamic manner` in Section 6.2), one can observe Co$^2$L adopts a much stronger backbone (ResNet-18) than ours  (two fully connected layers) when experimenting on R-MNIST. We argue that the stronger backbone (feature extractor) in Co$^2$L is the main reason for its higher performance than us on R-MNIST. To verify this point, we simply replace the backbone of GPE with ResNet18 without tuning hyperparameters. The results rise to 98%, which is comparable with Co$^2$L. Note since Co$^2$L is not fully open-sourced, we cannot get access to its detailed training settings, such as the replay schemes, knowing which may further improve our results. In summary, we highlight **two advantages** of GPE over Co$^2$L:
> >   -  GPE does not need memory buffer in design, while Co$^2$L depends on memory buffer to perform well. Therefore, GPE is more suitable for the task of incremental VHD since storing videos is memory consuming;
> >   -  GPE gets rid of complicated and sensitive hyper-parameters tuning while Co$^2$L needs to adjust the distillation temperatures and distillation power in each experiment. Thus, GPE is easier to use in practice and more scalable to various tasks.
> >
> > [Ref1] Constrained Few-shot Class-incremental Learning, Hersche et al. (CVPR 2022)
> >
> > [Ref2] Prototype Augmentation and Self-Supervision for Incremental Learning, Zhu et al. (CVPR 2021)
> >
> > [Ref3] Co$^2$L: Contrastive Continual Learning, Cha et al. (ICCV 2021)

---

> > > ### Comment · Reviewer_6gVY · 2022-11-19
> > > **Follow-up questions/comments**
> > >
> > > Thank you for the clarification. My concerns regarding W1, W2, W4, and W5 were adequately addressed.
> > > Some other concerns:
> > > 1) In A.1 of the updated appendix, it is mentioned that no video is stored locally. I do not believe this is a good idea. The entire dataset consists only of 5100 videos, and I strongly believe that all the videos should be maintained by the dataset organizer. If the dataset is a million-scale, it makes total sense to provide the video links. However, it has only 5100 videos, so every video should matter. Providing only the video links for this small-scale dataset does not seem like a good decision. This can cause inconsistent usage across different users in the long term creating different outcomes or unfair comparisons. A good example is ActivityNet; the organizer originally provided only the video links, but users requested the full dataset because there were inconsistent results/comparisons so the organizer finally started to provide the whole dataset after a few years.
> > > 2) In A.5, the authors argue that other advanced incremental learning techniques including Co2L cannot be well applied to complicated applications such as incremental VHD. I cannot fully agree with this point. Could you elaborate on this?
> > > 3)  In A.6, it is mentioned that the goal of GPE is to reduce the discrepancy across different domains. However, I still wonder how the learned prototypes of T1-T4 can remain in the same cluster across different training stages in Figure 13.

---

> > > > ### Author Response · Authors · 2022-11-19
> > > > **Response to the remaining concerns.**
> > > >
> > > > We are pleased to receive your further comments and suggestions! Please see our answers to your questions below.
> > > > - **About the video links (Q1).** Thanks for your suggestions. Due to legal requirements, we cannot host the videos locally and release them to external users in the pattern of dataset. However, since all of these videos are publicly available on video platforms and everyone is free to download and store them on their local devices, the dataset can definitely be ensured to be consistent and trackable. **We will take necessary measures to maintain the dataset such that each user can get access to a stable and consistent dataset** (e.g., work with 3rd-party personnel or organizations). In summary, the video link issue is not a problem and the integrity of dataset is guaranteed.
> > > > - **Comparison of practicability among methods (Q2).** GPE is more suited to practical scenarios due to three reasons.
> > > >   - Replay-based techniques are used in DER and Co$^2$L to decide which image/video to be preserved in the memory buffer. However, selecting videos is much more difficult than selecting images. Taking the "herding" scheme as an example. This selection method calculates the mean representation of each domain, and then picks the top K nearest neighbors' corresponding videos as the candidates. This operation introduces non-negligible costs in video inference since all videos have to be inferred. Adding that each video usually contains multiple domains, the selection process gets more difficult due to processing video overlaps. **By design, GPE does not need replay schemes, thus being free from above problems.** However, Methods such as DER and Co2L fail to perform well without using the replay scheme.
> > > >   - Co$^2$L needs a data augmentation module to boost performance. We argue that **it is untrivial to employ data augmentation schemes for videos.** Apart from the elaborate augmentation scheme needed to design for stable training with videos, it also incurs large computational costs by applying augmentation to video frames, thus hindering applications in practical scenarios.
> > > >   - **GPE gets rid of laborious hyper-parameters tuning process.** In contrast, Co$^2$L introduces many hyper-parameters in its distillation module, including the distillation temperatures and powers.
> > > > Due to above advantages, GPE is more suitable than other methods to deal with incremental video highlight detection in practical usage.
> > > > - **About the learned prototypes (Q3).** The main reason of why prototypes do not form clusters is **we model the learning task in each training stage as a binary classification problem (highlight versus negative prototypes).** Therefore, positive domains are all regarded to belong to the same class of "highlight" and there is no need to discriminate their domain labels. Based on above explanation, the small discrepancy among prototypes from different domains as shown in `Figure 14` just corroborate our purpose.  Below is an example for further clarification. Suppose we want to distinguish the dogs and cats (analogous to the highlight/negative classes), each species of dog/cat is recognized as a separate domain (analogous to "eating", "representation", etc.)  Since we only want to tell whether an input image belongs to dogs or cats (i.e., highlight or negative) regardless of their species, the learned prototypes shall show small differences among varied species, but have large inter-class distances. We hope our answer addresses your concerns.

---

> ### Author Response · Authors · 2022-11-26
> **The latest response from the authors.**
>
> Dear reviewer `6gVY`:
>
> We have replied to your concerns in our responses. The updated manuscript with a detailed `Appendix` is also available for your reference. Please take a look and feel free to raise any further concerns/questions. If no more questions, we would greatly appreciate it if you could adjust the rating. Thanks!

---

### Official Review · Reviewer_nvTS · 2022-10-23

**Confidence:** 3
**Clarity, Quality, Novelty And Reproducibility:** The work is mostly novel.
**Correctness:** 3
**Technical Novelty And Significance:** 3
**Empirical Novelty And Significance:** 3
**Recommendation:** 6

**Strength And Weaknesses:**

Strength: Most parts are explained clearly for the equations.Most of the procedures seems reasonable.

Weakness:
1) How to learn the two groups of trainable prototypes by equation (3)? Do the  two groups of trainable prototypes are trained by 2 class clasiification by equation 3. More explanation should be given for k*m before equation (1) and how the k*m matrix is used in Figure 4. The distance based clustering of Figure 4 is not mentioned in the paragraphs.
2) What does it mean by Average the metrics of current stage and Average the metrics of all stages in Algorithm 1.No explanation.

**Summary Of The Paper:**

Most of the paper is clearly written. A new dataset for incremental learning is provided. The fixed and dynamic modes of Global Prototype Encoding is proposed and outperform the state-of-arts.

**Summary Of The Review:**

 It offers a new dataset and a new method for future research in its direction, which sounds reasonable. The details of the protype learning should be explained more clearly.

---

> ### Author Response · Authors · 2022-11-08
> **Explanations of the training procedure and the evaluation protocol.**
>
> Thanks for the valuable comments. We are sorry for the confusion caused. We have rephrased/ reorganized relative contents in the latest manuscript to make it clearer. Please see below for our answers.
> - **About the training procedure.** The training procedure of prototypes is carried out as a **binary classification problem**. Supposing that we have the highlight prototypes $H$ and negative prototypes $V$, both of which are in the shape of $k\times m$ where $k$ is the number of prototypes and $m$ is the dimensionality of features $f$ outputted by the transformer. The L2 distance ($d_H$) between the output feature $f$ and $H$ is calculated using `Eq.(1)`. Similarly, we obtain the distance between $f$ and $V$, i.e., $d_V$. Having $d_H$ and $d_V$, we can transform them into the probability fashion using `Eq.(2)` for calculating cross-entropy loss. In `Figure 4`, each row (length is $m$) of the prototype matrix is depicted as an individual marker, whose color represents vanilla (blue series) or highlight (red series) prototypes. During the optimization of GPE for incremental learning, prototypes are only allowed to change within a small range, thus helping reserve knowledge learned in previous stages. I hope the above explanations may help you better understand the training pipeline of GPE.
>
> - **About the averaged metrics.** During each training stage (task), we report the evaluation results on the testing set as stage-wise results. As described in `Section 3` (Problem Statement), the testing set may contain more than one video highlight domain, and we only perform evaluation on domains that are involved in current and previous training stages.  Specifically, in stage-wise testing, we calculate the mAP of each domain, then average them to get the "averaged metric of the current stage". When all of the training stages are complete, we report the overall results by averaging the stage-wise results, getting the "averaged metric of all stages". In `Table 3`, the top four rows are the "averaged metrics of the current stage", each corresponding to one of four stages, and the bottom row represents the "averaged metric of all stages".

---

> ### Author Response · Authors · 2022-11-26
> **The latest response from the authors.**
>
> Dear reviewer `nvTS`:
>
> We have replied to your concerns in our responses. The updated manuscript with a detailed `Appendix` is also available for your reference. Please take a look and feel free to raise any further concerns/questions. If no more questions, we would greatly appreciate it if you could adjust the rating. Thanks!

---

### Official Review · Reviewer_LRmd · 2022-10-28

**Confidence:** 4
**Correctness:** 3
**Technical Novelty And Significance:** 4
**Empirical Novelty And Significance:** 4
**Recommendation:** 6

**Clarity, Quality, Novelty And Reproducibility:**

Novelty:

++ The task formulation and the proposed solution are concise, convincing, and novel. A seemingly reasonable approach has been proposed in this manuscript for the VHD. Compared to the prior work and baseline introducing GPE it achieves competitive results.

Clarity:

++ Some of the manuscript sections have been written exceptionally well whereas there has been clear confusion in the dataset section. However, a brief insight into the problem and background information has been provided which is necessary to understand the major issue in the prior work.
++ The manuscript also clearly describes the improvements and adequately contextualizes the contributions in such a way that it makes a good starting point for a novice reader.

Reproducibility:

++ The author has agreed to release the code and dataset made publicly available in some point of time in the future. However, looking at the algorithmic prototype and architecture, it can be reproducible at some point.

**Strength And Weaknesses:**

Strengths:

++ The proposal of the GPE is incremental and end-to-end. The main advantage of GPE is that it can identify the domains via learning vanilla prototypes, whereas prior approaches fail for the same as the fixed number of highlight categories have been defined before learning. It can also be detected in the experiments which have been performed on the LiveFood dataset by improving the performance by a good margin.

Weaknesses:

-- After reading the manuscript and performing research on this there have some concerns my mind which is stated below:
The language for the use of the dataset is quite unclear in the entire manuscript, as from the first point-of-view, it looks like the author has curated/created the dataset, whereas moving ahead the dataset has been collected from some XYZ sources, and there has been no direction that from which settings/how the dataset has been curated/collected. Could the author clarify this more if I have missed something in the manuscript for the same?

-- In regards to the dataset statistics, I believe the dataset does not have larger statistical effects as the average video lengths or the total number of videos are less compared to the open datasets as mentioned in Table 2. Can I ask the authors if they can give clear direction on why they need to propose a dataset?

**Summary Of The Paper:**

The paper addresses a crucial problem in computer vision: Video Highlights Detection (VHD) which aims to target the appealing domains for the given video(s). More general approaches are based on world assumptions which lead to poor scalability and to address this issue the author proposes a new method called Global Prototype Encoding (GPE) which learns incrementally for adaptation of the new domains. Additionally, the author also uses the LiveFood dataset which is a finely annotated dataset for the experiments. Together with strong and exhaustive experimental results, the authors showcase the practical usability and competitive performance against the prior work for specific LiveFood datasets.

**Summary Of The Review:**

Overall, currently at this stage, this is a good manuscript. I like the simplicity and wide applicability of the proposed GPE approach. There are some concerns which has been raised in the weakness section regarding the dataset proposal which holds me with the current rating, however, if given strong reasons and clarity, I'm happy to increase the rating for the same. A detailed literature review, a complete overview of each component, and detailed experiments and ablation studies help to give a good insight into the manuscript.

---

> ### Author Response · Authors · 2022-11-08
> **Response to concerns on dataset**
>
> We highly appreciate your valuable comments and suggestions. Below are the answers to each question. Besides, we also revised the manuscript with added `Appendix`, explaining more details of the dataset and experiments. Please refer to Appendix in the latest manuscript for more information.
> - **Basic information of the dataset.** We are sorry that more details about the dataset are omitted in the draft due to space limits. Please refer to `Appendix A.1/2/3/4` in the latest manuscript for details on building the dataset, including privacy preservation, visualization, quality control and the annotation tool. For your convenience, here we provide brief answers to your questions. Please check the Appendix for the more complete version.
>
>   - **Raw data collection.** We first collect raw videos that are published in a selected period from popular video-sharing platforms (e.g. https://www.douyin.com/). **All videos have been authorized by creators to view and share before going public.** For getting food-related videos, we use keywords like "gourmet" or "food" for retrieving videos on the platform.
>   - **Data filtering and annotation.** The raw collection still contains a lot of noisy and inappropriate videos. To get a high-quality dataset, the annotators are asked to review all videos to filter out those undesired. During this process, we employ an in-house tool that is specifically developed for annotation (cf. `Appendix A.4`). The tool supports playing videos at user-given speed and displaying video frames with adjustable fps. Annotators first select a rough time range of a highlight clip after viewing the whole video. Then the tool shows all frames extracted in 3fps in that time range so as to assist annotators to decide the exact start and end timestamps of highlights. Strict quality control (cf. `Appendix A.3`) is imposed to guarantee the quality of LiveFood.
> - **The advantages of LiveFood over other datasets.** To the best of our knowledge, **LiveFood is the first dataset providing high-quality category or domain annotations for video highlight detection (VHD)**, thus offering a desirable benchmark for future research on incremental VHD. As explained in `Section 4`, there are two major drawbacks in previous datasets. **First**, they all lack fine-grained domain labels as in LiveFood, therefore inapplicable for our concerned task of incremental VHD. **Second**, due to the high expenses of annotations and filtering, larger datasets often lack reliable quality control, leading to inferior dataset quality and thus adversely affecting the evaluations of VHD methods. For example, previous datasets such as PHD and Video2GIF (see Table 2 for their statistics) resort to the GIF creation website instead of annotators for getting the ground truth of video highlights: users upload a video and create a GIF clip website, then the video and GIF are regarded raw video and corresponding highlight respectively. Obviously, though larger than LiveFood, their quality is much worse than ours which applies strict quality control during the manual annotation process. We also believe our high-quality dataset can largely facilitate a wide range of research, not only on incremental learning and VHD but also on general video understanding, e.g. video moment localization (by utilizing the start/end time points of highlight clips).
> - **Reproducibility.** For easy reproduction of our results, we upload the source code in zipped files to the `supplementary material`. One may run the code directly to exactly reproduce the reported results on R-MNIST (cf. `Table 4` and `Appendix Table 7`).

---

> ### Author Response · Authors · 2022-11-26
> **The latest response from the authors.**
>
> Dear reviewer `LRmd`:
>
> We have replied to your concerns in our responses. The updated manuscript with a detailed `Appendix` is also available for your reference. Please take a look and feel free to raise any further concerns/questions. If no more questions, we would greatly appreciate it if you could adjust the rating. Thanks!

---

### Author Response · Authors · 2022-11-18
**Response to all**

We sincerely appreciate each reviewer's time and efforts in reviewing our manuscript. **We have updated the manuscript to take in all the reviewers' comments/suggestions**. Apart from revisions in the main text, we attach `Appendix` after the `Reference` for your reference, which contains **more dataset-related information**, **visualization of results**, **methodology analysis**, etc.  Please feel free to check the latest manuscript for more details.

---

### Decision · Program_Chairs · 2023-01-20

**Decision:**

Reject

**Justification For Why Not Higher Score:**

The two major concerns are critical enough for rejection.

**Justification For Why Not Lower Score:**

N/A

**Metareview: Summary, Strengths And Weaknesses:**

The paper claims three main contributions: 1) new task (incremental video highlight detection), 2) new dataset (LiveFood dataset, which contains 5100 food related videos labeled with 4 "domain" categories), 3) a prototypical learning approach that is effective on the proposed task & dataset. The reviewers acknowledged that the proposed task is interesting and practical, and that the domain labels provided in the newly collected dataset could be useful to the community.

However, there were concerns about the availabilities of the dataset -- the videos will be released in the form of URLs due to the copyright concerns. This itself could be OK if the dataset was large (in the millions), but at just about 5K videos a few missing videos can lead to a large swing in the performance. The authors assured that they will make every effort to make the videos available, but given that the authors do not retain the copyright the reviewers were unconvinced whether the videos will truly remain available for a potential long-lasting impact.

There was a critical concern on the limited technical novelty. The proposed approach is a combination of well-known backbones (ConvNeXt and transformers) and learning techniques (prototypical learning). Especially, prototypical learning has been studied extensively in the incremental learning literature, which is what this paper is focusing on (although in a specific application domain of video highlight detection, which hasn't been studied extensively before).

The authors provided a rebuttal, addressing most of the minor concerns raised by the reviewers, but it didn't successfully address the two core concerns above (data availability and technical novelty).

After carefully reading the reviews, the rebuttal, and the discussion threads, this meta-reviewer did not find strong reasons to accept the paper despite the lingering concerns on novelty and the dataset availability, which are two of the three claimed contributions in this paper. Given this, we are recommending rejection at this time.